## PERSPECTIVE

### Chronic hypoxia-induced carotid body hypertonicity: Establishing 'base cAMP' in the search for therapies

Ken D. O'Halloran[1] ,
Rodrigo Iturriaga[2] 
and Asuncion Rocher[3]

[1]*Department of Physiology, University College Cork, Cork, Ireland*
[2]*Instituto de Ciencias Biomédicas, Universidad Autónoma de Chile, Chile*
[3]*Instituto de Biomedicina y Genética Molecular (IBGM), CSIC – Universidad de Valladolid, Valladolid, Spain*

Email: k.ohalloran@ucc.ie

Handling Editors: Kim Barrett & Harold Schultz

The peer review history is available in the Supporting Information section of this article (https://doi.org/10.1113/JP290429#support-information-section).

Cardiorespiratory and metabolic diseases are characterised by enhanced carotid body (CB) chemosensory activity, which contributes to autonomic and cardiovascular morbidity (Iturriaga, 2023; Shen & Paton, 2025). There is intense research focus on the underlying mechanisms. Importantly, resection of the CBs in animal models of disease most often resolves cardiometabolic perturbations arising from sympathetic hyperactivity linked to CB hypertonicity (Iturriaga, 2023; Sacramento et al., 2017). However, this approach is not a viable solution for the treatment of human disease because reflex responses to hypoxia (and many other signals) are lost; therefore, the intervention is reasoned on balance to be potentially unsafe. As such, there is keen interest in the search for device- or drug-driven effective targeting of the CBs as a therapeutic strategy. Ideally, an intervention would reverse disease-related CB sensitisation, at the same time as maintaining basal discharge of the chemosensory organ with retention of appropriate chemoreflex responsiveness.

Chronic hypoxia (CH), a feature of respiratory diseases, causes CB hypertonicity and sensitisation to hypoxia. In this issue of *The Journal of Physiology*, Nathanael et al. (2026) perform a stylish suite of studies to explore fundamental questions pertaining to CH-dependent CB plasticity of translational value relevant to human disease. The rationale for the study stems from prior work by the Birmingham group revealing an essential role for CD73, an ecto-5′-nucleotidase that converts AMP to adenosine, which excites the CB, in CB function *per se* (Holmes et al., 2018). The primary questions addressed were: does CH increase basal single chemoafferent discharge frequency and potentiate responses to acute hypoxia? Is there a pivotal role for CD73 in hypoxia-induced CB hypertonicity?

In isolated superfused CBs of rats exposed to hypoxia for 9–10 days, basal discharge and hypoxic responses of CB single chemoafferent fibres were increased demonstrating intrinsic sensitisation of CB glomus cells in response to sustained exposure to hypoxia. CD73 was expressed in mature glomus (type 1) cells of the CB (colocalised with tyrosine hydroxylase). Chronic hypoxia increased the density of CD73-positive type I cells of the CB but did not affect whole CB size, which probably relates to the relatively mild model of CH used in the study, in terms of intensity and duration of exposure (12% inspired oxygen for 9–10 days). AOPCP, an inhibitor of CD73, produced dose-dependent, rapidly reversible decreases in CB chemosensory discharge in normoxic and CH-exposed CBs. The data are consistent with a role for CD73 in mediating the CB hyperactivity but stop short of demonstrating it definitively. Adenosine was excitatory to CB discharge and further elevated discharge frequency in the hyperactive CH-exposed CBs, but sensitivity to adenosine was not increased in CH-exposed CBs. This could be viewed as detracting somewhat from the suggestion that CD73 drives CB hypertonicity, which could ostensibly be mediated by several other proposed mechanisms, but importantly demonstrates that CD73 is a viable target to offset CB hyperactivity. In support of the hypothesis, however, it is also plausible that a greater pool of extracellular adenosine is maintained in CH-exposed CBs facilitating enhanced signalling. Future experiments centred on addressing this open question will be important to address. AOPCP also decreased the exaggerated hypoxic responsiveness of CH-exposed CB preparations. Again, this does not explicitly provide for an obligatory mechanism but significantly establishes CD73 as an effective target. Importantly, exaggerated responses to hypoxia in CH-exposed CBs were normalised to control in the presence of AOPCP. Similarly and importantly, SQ22536, a transmembrane adenylyl cyclase inhibitor, decreased basal CB discharge in normoxic and CH-exposed CBs and restored hypoxic sensitivity to normal levels in CH-exposed CBs. The results point to a pivotal role for cAMP and downstream effectors.

Complementary studies performed in anaesthetised rats revealed that AOPCP causes dose-dependent decreases in arterial blood pressure mediated predominantly via the CB because the effect was attenuated following carotid sinus nerve section. Interestingly, reflex responses to severe hypoxia were preserved in the presence of AOPCP, whereas these were profoundly lost following carotid sinus nerve sectioning in control and CH-exposed animals. The capacity for CD73 blockade to alter integrated cardiorespiratory responses, especially exaggerated responses in CH-exposed animals is an important advance in the search for therapies, but the observations need next to be demonstrated in freely behaving animals without the confounding influence of anaesthesia. Moreover, CH-exposed rats did not develop hypertension. It would be most interesting to determine whether CD73 blockade mitigates hypertension associated with CB hyperactivity across a range of available animal models.

That CB and integrated cardiorespiratory responses to severe hypoxia can be elicited even in the presence of high-dose AOPCP is not only important in the context of safety in disease treatment, but also reveals that the abrogation of CB hyperactivity is not merely a result of blockade of a critical distal node in hypoxic chemotransduction, which would have limitations as a therapy as per concerns associated with CB lesion. Collectively, the results are consistent with a role for adenosine in setting CB tonic chemosensory activity and the CB response to mild-to-moderate hypoxia. This is consistent with the hypothesis that the adenosine cascade may indeed set a higher gain in CB discharge following CH exposure.

Pharmacotherapy offers a potentially promising approach to tackle CB

The Journal of Physiology

hypertonicity, which presents in a broad range of cardiorespiratory, autonomic and metabolic disease states. Fundamental studies revealing the mechanisms contributing to CB sensitisation culminating in the identification of druggable targets that can safely be trialled in humans are centre stage. Nathanael et al. (2026) provide an intriguing study seeking to delineate the mechanism of CB sensitisation to CH. Pharmacological blockade of CD73 implicates the adenosine pathway in CH-induced CB hyperactivity but also establishes 'base cAMP' for further exploration of the efficacy of targeting this pathway in other disease states. Targeted delivery to the CB remains an obvious challenge but one step at a time!

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

## Additional information

### Competing interests

No competing interests declared.

## Author contributions

K.O.H., R.I. and A.R. were responsible for the conception or design of the work; drafting the work or revising it critically for important intellectual content; and giving final approval of the manuscript submitted for publication. All authors agree to be accountable for all aspects of the work.

## Funding

No funding was received.

## Keywords

adenosine, carotid body, chronic hypoxia, hypertonicity

## Supporting information

Additional supporting information can be found online in the Supporting Information section at the end of the HTML view of the article. Supporting information files available:

**Peer Review History**

