## [Peer Review History · The Journal of Physiology]

Chronic hypoxia-induced carotid body hypertonicity: establishing 'base cAMP' in the search for therapies

Ken D O'Halloran, Rodrigo Iturriaga, and ASUNCION ROCHER
DOI: 10.1113/JP290429

Corresponding author(s): Ken O'Halloran (k.ohalloran@ucc.ie)

The following individual(s) involved in review of this submission have agreed to reveal their identity: Andrew P Holmes (Referee #1)

Review Timeline:

Submission Date:	06-Nov-2025
Editorial Decision:	18-Nov-2025
Revision Received:	19-Nov-2025
Accepted:	20-Nov-2025

Senior Editor: Kim Barrett

Reviewing Editor: Harold Schultz

Transaction Report:

Dear Dr O'Halloran,

Re: JP-P-2025-290429 "**Chronic hypoxia-induced carotid body hypertonicity: establishing 'base cAMP' in the search for therapies**" by Ken D O'Halloran, Rodrigo Iturriaga, and ASUNCION ROCHER

Thank you for submitting your manuscript to The Journal of Physiology. It has been assessed by a Reviewing Editor and by 1 expert referee and we are pleased to tell you that it is acceptable for publication following satisfactory revision.

The review comments are copied at the end of this email.

Please address all the points raised and incorporate all requested revisions or explain in your Response to Referees why a change has not been made. We hope you will find the comments helpful and that you will be able to return your revised manuscript within 2 weeks. If you require longer than this, please contact journal staff: jp@physoc.org.

REVISION CHECKLIST:

We look forward to receiving your revised submission.

Yours sincerely,

Kim Barrett
Senior Editor
The Journal of Physiology

REQUIRED ITEMS

1) - The reference list must be in alphabetical order, rather than numbered, to comply with our Journal format.

EDITOR COMMENTS

Reviewing Editor:

Thank you for submitting this well-written invited perspective article for consideration to the Journal of Physiology. The article has been reviewed by the focus authors, who have a couple of minor comments. The authors' clarifications may or may not alter your position, but should be addressed. Please resubmit the article with your response and any revisions.

REFEREE COMMENTS

Referee #1:

Thank you very much for taking the time to put together this lively and thought provoking perspectives manuscript. I enjoyed reading it very much. The manuscript accurately describes the associated focus paper and does an excellent job of emphasising many of the key findings. It also provides some excellent insight into the current state of the field and its future direction. I agree with all the future suggestions especially regarding targeting CD73 in hypertension models, and also in awake freely moving animals. This is a key next step towards translation. I only have a couple of minor comments/suggestions:

1. The title focuses on cAMP, and I agree that this is very important. However, cAMP is only briefly mentioned in the final paragraph. Perhaps the discussion of cAMP could be slightly expanded and possibly include that the focus article observed that SQ22536 attenuated CB hyperactivity in CH CBs similar to that of AOPCP, highlighting the importance of tmAC and cAMP?

2. We think that the relatively mild model of CH is probably why there wasn't significant CB hypertrophy. Perhaps the use of a milder model of CH used in the study (that still caused strong CB hyperactivity) can be mentioned?

Thank you again.

END OF COMMENTS

REQUIRED ITEMS

1) - The reference list must be in alphabetical order, rather than numbered, to comply with our Journal format.

RESPONSE: The reference list is now provided in alphabetical order.

EDITOR COMMENTS

Reviewing Editor:

Thank you for submitting this well-written invited perspective article for consideration to the Journal of Physiology. The article has been reviewed by the focus authors, who have a couple of minor comments. The authors' clarifications may or may not alter your position, but should be addressed. Please resubmit the article with your response and any revisions.

RESPONSE: Thank you for this helpful summary. We have addressed the referee's comments and have revised the text in accordance with some useful suggestions.

REFEREE COMMENTS

Referee #1:

Thank you very much for taking the time to put together this lively and thought provoking perspectives manuscript. I enjoyed reading it very much. The manuscript accurately describes the associated focus paper and does an excellent job of emphasising many of the key findings. It also provides some excellent insight into the current state of the field and its future direction. I agree with all the future suggestions especially regarding targeting CD73 in hypertension models, and also in awake freely moving animals. This is a key next step towards translation.

RESPONSE: Thank you for this enthusiastic response. We are glad the perspective was well received.

I only have a couple of minor comments/suggestions:

1. The title focuses on cAMP, and I agree that this is very important. However, cAMP is only briefly mentioned in the final paragraph. Perhaps the discussion of cAMP could be slightly expanded and possibly include that the focus article observed that SQ22536 attenuated CB hyperactivity in CH CBs similar to that of AOPCP, highlighting the importance of tmAC and cAMP?

Response: This is good suggestion, and we are grateful to you for pointing it out. We have now briefly expanded the text to draw focus to the effects of SQ22536, which provides further mechanistic insight, establishing the importance of cAMP and downstream effectors.

2. We think that the relatively mild model of CH is probably why there wasn't significant CB hypertrophy. Perhaps the use of a milder model of CH used in the study (that still caused strong CB hyperactivity) can be mentioned?

Response: We now add that the lack of change in CB size likely relates to the relatively mild model of CH used in the study, in terms of intensity and duration of exposure (12% inspired oxygen for 9-10 days).

Thank you again.

Response: Our distinct pleasure. The original paper is an important contribution to the literature.

END OF COMMENTS

Dear Professor O'Halloran,

Re: JP-P-2025-290429R1 "**Chronic hypoxia-induced carotid body hypertonicity: establishing 'base cAMP' in the search for therapies**" by Ken D O'Halloran, Rodrigo Iturriaga, and ASUNCION ROCHER

We are pleased to tell you that your paper has been accepted for publication in The Journal of Physiology.

Please note that Perspective articles are not typically covered by institutional open access agreements with our publisher, Wiley. Wiley do not offer article processing charge (APC) discounts for smaller article types in hybrid subscription journals, meaning that if you wish for your Perspective to be published Open Access, you will have to pay the full APC. As such, we recommend authors publish Perspectives 'behind the paywall', where they will become freely accessible after a 12-month embargo (i.e. please select the NON open access option via Wiley Author services during proofing).

Should you wish to pay for Open Access, you will be able to place an order by logging into Wiley Author services.

Yours sincerely,

Kim Barrett
Senior Editor
The Journal of Physiology

IMPORTANT POINTS TO NOTE FOLLOWING ACCEPTANCE OF YOUR PAPER:

- **IMPORTANT NOTICE ABOUT OPEN ACCESS:** To assist authors whose funding agencies mandate immediate public access to published research findings, The Journal of Physiology allows authors to pay an Open Access (OA) fee to have their papers made freely available immediately on publication.

- You can help your research get the attention it deserves! Check out Wiley's free Promotion Guide for best-practice recommendations for promoting your work at: www.wileyauthors.com/eoo/guide. You can learn more about Wiley Editing Services which offers professional video, design, and writing services to create shareable video abstracts, infographics, conference posters, lay summaries, and research news stories for your research at: www.wileyauthors.com/eoo/promotion.

- If you would like to receive our 'Research Roundup', a monthly newsletter highlighting the cutting-edge research published in The Physiological Society's family of journals (The Journal of Physiology, Experimental Physiology, Physiological Reports, The Journal of Nutritional Physiology and The Journal of Precision Medicine: Health and Disease), please click this link, fill in your name and email address and select 'Research Roundup':

<https://www.physoc.org/journals-and-media/membernews>

EDITOR COMMENTS

Reviewing Editor:

Wonderful communication and revision.